# Progression of Obstructive Sleep Apnea Syndrome in Pediatric Patients with Prader–Willi Syndrome

**DOI:** 10.3390/children9060912

**Published:** 2022-06-17

**Authors:** Shi-Bing Wong, Mei-Chen Yang, I-Shiang Tzeng, Wen-Hsin Tsai, Chou-Chin Lan, Li-Ping Tsai

**Affiliations:** 1Department of Pediatrics, Taipei Tzu Chi Hospital, Buddhist Tzu Chi Medical Foundation, New Taipei City 23142, Taiwan; ybinwang@gmail.com (S.-B.W.); tsaiwh.tw@gmail.com (W.-H.T.); 2School of Medicine, Tzu Chi University, Hualien 97004, Taiwan; mimimai3461@gmail.com (M.-C.Y.); bluescopy@yahoo.com.tw (C.-C.L.); 3Division of Pulmonary Medicine, Department of Internal Medicine, Taipei Tzu Chi Hospital, Buddhist Tzu Chi Medical Foundation, New Taipei City 23142, Taiwan; 4Department of Research, Taipei Tzu Chi Hospital, Buddhist Tzu Chi Medical Foundation, New Taipei City 23142, Taiwan; istzeng@gmail.com

**Keywords:** Prader–Willi syndrome, obstructive sleep apnea syndrome, sleep-disordered breathing, polysomnography

## Abstract

Obstructive sleep apnea syndrome (OSAS) is one of the most common comorbidities in patients with Prader–Willi syndrome (PWS) and causes significant consequences. This observational study was conducted to investigate the progression of OSAS in pediatric patients with PWS, who had not undergone upper airway surgery, through a longitudinal follow-up of their annual polysomnography results. Annual body mass index (BMI), BMI z-score, sleep efficiency and stages, central apnea index (CAI), obstructive apnea–hypopnea index (OAHI), and oxygen saturation nadir values were longitudinally analyzed. At enrollment, of 22 patients (10 boys and 12 girls) aged 11.7 ± 3.9 years, 20 had OSAS. During the 4-year follow-up, only two patients had a spontaneous resolution of OSAS. The average BMI and BMI z-score increased gradually, but CAI and OAHI showed no significant differences. After statistical adjustment for sex, age, genotype, growth hormone use, and BMI z-score, OAHI was associated with the BMI z-score and deletion genotype. In conclusion, OSAS is common in patients with PWS, and rarely resolved spontaneously. Watchful waiting may not be the best OSAS management strategy. Weight maintenance and careful selection of surgical candidates are important for OSAS treatment in patients with PWS.

## 1. Introduction

Prader–Willi syndrome (PWS) is a rare genetic disorder with an estimated prevalence of 1/10,000–1/30,000 individuals; the loss of function of the paternal chromosome 15q11-13 affects multiple systems [1]. Sleep-disordered breathing (SDB) is one of the most common comorbidities of PWS and causes significant consequences for patients with PWS [2,3]. SDB includes central apnea (CA) and obstructive sleep apnea (OSA). CA during sleep is a prominent feature in infants with PWS, and although it can gradually improve with maturation, it can also cause unexpected death [4,5,6]. Oxygen therapy may significantly reduce the CA index (CAI) in infants with PWS [5]. However, OSA frequently occurs in older children with PWS, and its risk factors include obesity and obesity-related narrowing of the upper airways, facial dysmorphism, increased viscosity of respiratory secretions, scoliosis, and hypotonia [2,3,7]. Furthermore, growth hormone (GH) therapy for PWS-related endocrinopathies may exacerbate obstructive sleep apnea syndrome (OSAS) [8,9].

Long-term OSAS may worsen patients’ behavior problems, cognitive deficits, and cardiovascular and endocrine disorders [10,11]. Treatment options include behavioral intervention, weight control, continuous positive airway pressure (CPAP), oral appliance therapy, and surgery such as adenotonsillectomy or other orthognathic surgeries [12,13]. Nevertheless, in patients with PWS, it is difficult to obtain complete resolution of sleep apnea through adenotonsillectomy alone. In some children with PWS, an increase in CA can even occur postoperatively [14,15]. Other orthognathic surgeries, including maxillomandibular advancement, are rarely performed in patients with PWS [16]. CPAP treatment improves sleep and quality of life in patients with severe OSAS, but side effects are frequently encountered [17]. Mask- or pressure-related discomfort, nasal congestion, headache, and bloating are common complaints among patients undergoing CPAP treatment [18]. Consequently, some patients refuse to undergo surgery or CPAP treatment after the diagnosis of OSAS. In the case of neurotypical children, some would have spontaneous resolution over time, and watchful waiting is an acceptable management for those children with mild or moderate OSAS [19]. However, there is a lack of data on the long-term progression of OSAS in patients with PWS that can help us determine an appropriate management strategy. Therefore, we conducted this study to evaluate the progression of OSAS using consecutive polysomnography (PSG) data obtained from pediatric patients with PWS in our institution.

## 2. Materials and Methods

### 2.1. Patients

In our institution, patients with PWS were annually screened for OSAS using PSG. We retrospectively reviewed their PSG reports from 2009 to 2017 and included 48 patients. All patients were diagnosed after one or more of the following tests: PWS-specific methylation-specific polymerase chain reaction, fluorescence in situ hybridization analysis, and microsatellite studies [20]. Among the 48 patients, 2 received CPAP treatment, 1 received adenotonsillectomy, and 23 had only 1- or 2-year longitudinal data. Therefore, 22 patients (84 PSG studies) who underwent at least 3 years of follow-up were included in this study. Among them, 18 patients were treated with GH, according to the protocol of de Lind van et al. [21]. This study was conducted in accordance with the Declaration of Helsinki and was approved by the local ethics committee of Taipei Tzu Chi General Hospital (07-XD02-003, approved at 12 March 2018). Written informed consent was waived because the study was a retrospective data analysis.

### 2.2. Overnight PSG and Sleep Variables

The annual overnight PSG for all patients was conducted at Taipei Tzu Chi Hospital. The PSG study included an electroencephalogram (F3, F4, C3, C4, O1, O2), electrooculogram, chin and leg electromyogram, and electrocardiography (modified V2 lead). Respiration was measured using nasal thermistors and respiratory inductance plethysmography belts on the chest and upper abdominal walls. Oxyhemoglobin saturation was measured using pulse oximetry. Data in each sleep stage were manually rated according to the guidelines of the American Academy of Sleep Medicine on the basis of 30-s epochs [22]. OSAS and hypopnea were defined as a near absence (≥90%) or a baseline decrease (≥30%) in the amplitude of ventilation for at least two breaths as measured by nasal thermistors and associated with 4% oxygen desaturation. Baseline was defined as the mean amplitude of stable breathing and oxygenation in 2 min preceding the event onset (in individuals with a stable breathing pattern during sleep) or that of the three largest breaths in 2 min preceding the event onset (in individuals without a stable breathing pattern). CA was defined as an absence of nasal airflow for ≥10 s without respiratory effort or shorter events associated with arousal, desaturation (≥3%), or bradycardia (≤60 beats per min) [22]. CAI was defined as the mean number of central apneic episodes per hour of sleep. The obstructive apnea–hypopnea index (OAHI) indicated the mean number of obstructive hypopneic and apneic episodes per hour of sleep. OSAS categories were defined according to OAHI (no OSAS, OAHI < 1; mild OSAS, OAHI ≥1–<5; moderate OSAS, OAHI ≥5–<10; and severe OSAS, OAHI ≥ 10) [23].

### 2.3. Normalized Body Mass Index (BMI) Z-Score

To adjust for age and sex, we converted patients’ BMI values into z-scores using age- and sex-specific BMI norms for Taiwan [24]. Each z-score was calculated by subtracting a patient’s BMI from the mean BMI for the patient’s age group and dividing the difference by the standard deviation (SD) of BMI for that age group. Overweight was defined as a BMI z-score of >1, and obesity was defined as a BMI z-score of >2.

### 2.4. Statistical Analysis

Statistical analyses were conducted using SPSS version 19.0 for Windows (IBM, Chicago, IL, USA). Descriptive data are presented as the mean ± SD. Repeated measures analysis of variance (ANOVA) was used to compare the annual CAI, OAHI, BMI, and BMI z-scores of individual patients. The Mann–Whitney U test and Spearman’s correlation were used to evaluate the relationships between OAHI, BMI, and BMI z-scores in the first year of enrollment. Repeated measures linear regression (an exchangeable working within-subject correlation model by generalized estimating equation (GEE)) was used to compute the average rates of progression of OAHI in male and female patients with PWS after adjusting for age and BMI z-scores. The longitudinal progressions of OSAS severity in patients with PWS during the 4-year follow-up period were evaluated by entering the interaction terms (gender × PSG time) into the GEE models. The coefficients of the interaction terms reflected how the rate of progression of OAHI differed according to gender among the patients. Two-sided *p* values of <0.05 were considered to be statistically significant.

## 3. Results

The demographic data and summary of polysomnographic variables during the longitudinal 4-year follow-up of patients with PWS are shown in Table 1. In total, 10 boys and 12 girls with PWS were enrolled in this study, and they underwent three or four consecutive PSG studies annually. The mean age at enrollment was 11.7 ± 3.9 years (boys, 13.3 ± 4.2 years; girls, 10.4 ± 3.3 years; *p* = 0.146). In this cohort, 17 patients had a paternal deletion of 15q11-13, and 5 patients were of the nondeletion type. At enrollment, their mean BMI was 22.8 ± 4.5 kg/m^2^, and the mean BMI z-score was 1.7 ± 1.3. Of the 22 patients, 15 (8 boys and 7 girls) were overweight or obese. The BMI z-score of both sexes was similar (boys, 2.2 ± 1.4; girls, 1.3 ± 1.0; *p* = 0.187). Of the 22 patients, 18 were treated with GH. Adrenal function had been tested in 20 patients and the results were all normal, including 16 patients with a peak cortisol greater than 500 nmol/liter in the insulin hypoglycemia test, and 4 patients with basal cortisol levels more than 280 nmol/liter. All 22 patients had normal thyroid function on annual follow-up. Regarding the PSG parameters, the CAI of the first PSG was 0.3 ± 0.5 per hour. The OAHI at the first year of enrollment was 5.7 ± 6.4 per hour (boys, 3.7 ± 2.6, girls, 7.4 ± 8.2; *p* = 0.19). Of the 22 patients, 20 (90.9%) had OSAS at enrollment, including 8 (36.4%) patients with mild, 11 (50%) patients with moderate, and 1 (4.5%) patient with severe OSAS. We found no correlation between BMI and OAHI (*p* = 0.93, Spearman’s correlation) or between BMI z-score and OAHI (*p* = 0.48, Spearman’s correlation).

During the consecutive 4-year follow-up, the OSAS severity showed a constant distribution. The last PSG follow-up revealed that 5.6% of patients had no OSAS, 38.9% had mild OSAS, 38.9% had moderate OSAS, and 16.7% had severe OSAS (Table 1). The OSAS severity of the 22 patients during the 4-year follow-up is illustrated in Figure 1. Based on the data from the last follow-up PSG, one patient with moderate OSAS (patient 7 in Figure 1) and one with mild OSAS (patient 8 in Figure 1) had a normalized OAHI score, indicating that OSAS barely resolved spontaneously in patients with PWS. These two female patients were aged 8 and 20 years at enrollment, respectively. The BMI z-score decreased from 3.4 to 2.1 in patient 7 and from 1.5 to 1.1 in patient 8. By contrast, two patients were initially normal (patient 3 and patient 5 in Figure 1) and developed OSAS during the longitudinal follow-up. Patient 3 was an 18-year-old male patient whose OAHI was 0.7/h at enrollment. During the three-year follow-up, the OAHI increased to 11.3/h. He did not receive GH therapy, and his BMI z-score increased from 2.9 to 3.8 in three years. Patient 5 was an 8-year-old boy whose OAHI increased from 0.1 to 9.7/h in three years. Although he received GH therapy, his BMI z-score still increased from 3.1 to 4.3 during the longitudinal follow-up. Accordingly, we presumed that obesity is an important risk factor for OSAS in patients with PWS.

However, the overall CAI (F = 1.09, *p* = 0.34, repeated measures ANOVA) and OAHI (F = 0.60, *p* = 0.52, repeated measures ANOVA) of patients with PWS did not differ significantly across the study period, despite that there was an increase in the obesity of patients with PWS annually. The mean BMI increased from 22.8 to 25.9 kg/m^2^ (F = 9.84; *p* = 0.002, repeated measures ANOVA), and the mean BMI z-score increased from 1.7 to 2.3 (F = 4.42; *p* = 0.03, repeated measures ANOVA). Because weight gain is a well-known risk factor for OSAS aggravation [25], there must be factors other than obesity affecting the progression of OSAS in patients with PWS.

Therefore, we applied a GEE model that adjusted for the patients’ variables, including BMI z-score, age, sex, PSG time, genotype, GH treatment, BMI z-score × PSG time, and genotype × PSG time, to evaluate the risk factors for OSAS progression in the consecutive 4-year follow-up (Table 2). The linear regression revealed that OAHI correlated positively with the BMI z-score and negatively with the non-Del genotype, significantly so. The regression coefficients of the BMI z-score and non-Del genotype were 4.88 (Wald χ^2^ = 16.85, *p* < 0.001) and −3.73 (Wald χ^2^ = 4.09, *p* = 0.043), respectively (Table 2). In this GEE model, PSG time, BMI z-score × PSG time, and genotype × PSG time revealed no significant regression coefficients with OAHI, indicating that the time domain (PSG time) in this study had no effects on OAHI changes, which further validated the conclusion that OSAS in patients with PWS rarely resolved spontaneously over time.

## 4. Discussion

We investigated the progression of OSAS in patients with PWS who had not undergone adenotonsillectomy. The prevalence of OSAS was high in these patients, and only a few patients showed spontaneous resolution during the 4-year PSG follow-up. After multivariate linear analysis, obesity and deletion genotype were identified as the risk factors for high OAHI.

PWS is a multisystem disorder caused by the loss of function of multiple genes from the paternal chromosome 15q11-13. Affected infants present with marked hypotonia since birth, which causes feeding difficulties and failure to thrive. The majority of patients exhibit delayed motor and language milestones, as well as intellectual disability [26]. SDB, including CA and OSAS, was frequently observed in patients with PWS. Studies have reported that infants and toddlers with PWS had irregular breathing and frequent CA, which were possibly associated with necdin deficiency, a universal character of PWS, resulting in abnormal central chemoreceptor sensitivity [27,28]. As a toddler, an excessive appetite develops, and the patients gradually become obese. Furthermore, facial dysmorphism, including micrognathia and small oropharynx, sticky secretions, and hypotonia, have been identified as risk factors for OSAS in patients with PWS [2]. Consistent with previous studies, we also detected a high prevalence of OSAS in patients with PWS in our study (90.9% at the time of enrollment). To our knowledge, this is the first study to provide longitudinal PSG data of patients without surgical or CPAP intervention, which is important to understand the natural course of OSAS in patients with PWS. During the consecutive 4-year follow-up, only two patients (9.1%) showed spontaneous resolution of OSAS. In contrast, for neurotypical children, it has been reported that 65% of patients with mild or moderate OSAS and 26% of those with severe OSAS underwent spontaneous resolution of OSAS [19]. Our result indicated that watchful waiting is possibly not the best strategy to manage OSAS in patients with PWS, and it is necessary to consider aggressive interventions, including adenotonsillectomy, CPAP, and oral appliance therapy, earlier in this special group of patients.

SDB would result in multiple neurocognitive and cardiovascular consequences, including deficits in general intelligence, mental flexibility, working memory, and heart ventricular functions [12]. In patients with PWS, SDB even contributes to increased mortality [29]. Nevertheless, studies have reported that patients with PWS undergoing adenotonsillectomy, the first-line treatment for OSAS in neurotypical children [12], have increased risk of developing postoperative complications and residual OSA [30,31]. As adenotonsillectomy is effective in some patients with PWS and OSAS, careful selection of appropriate surgical candidates is crucial. Drug-induced sedation endoscopy (DISE) is considered as a valuable tool when selecting patients with OSAS for adenotonsillectomy [32]. We have earlier reported the DISE findings of nine patients with PWS and OSAS, wherein multilevel obstruction was found in six patients (66.7%), and patients with partial or complete anterior–posterior tongue base collapse had a significantly higher AHI [33]. These findings may partially explain the inadequate efficacy of adenotonsillectomy in these patients because the mere removal of adenoids and tonsils would not resolve a multilevel airway obstruction and suggests that a procedure such as DISE, which allows for a better evaluation of factors contributing to OSAS, provides a better method for OSAS management.

In our study, obesity and deletion genotype were the two risk factors for OSAS in patients with PWS. In neurotypical children, obesity is a proved risk factor for OSAS [34,35]. Compared with nonobese children with OSAS, obese children also have a higher risk of postoperative residual OSA [36], whereas weight reduction significantly improves OSAS in obese children or teenagers, indicating that tonsil enlargement is not the major origin of OSAS in obese patients [37]. In our study, OAHI at enrollment showed no positive correlation with the BMI or BMI z-score of each patient. However, during the longitudinal follow-up, patients with higher BMI z-scores showed a higher risk of developing OSAS. In patients with PWS, progressive obesity was found after 30 months of age [38], and GH treatment has been demonstrated to stabilize the BMI increment [38,39]. Although most of our study patients were treated with GH, progressive obesity was observed in some patients. Close monitoring of food intake and adequate exercise are important for weight maintenance in patients with PWS [40] and are beneficial in the control of OSAS. Although no patients with PWS in this study had hypothyroidism or adrenal insufficiency, from the literature, 4–24% of patients of PWS would be comorbid with hypothyroidism [41,42], and 4.8–60% of patients with adrenal insufficiency [43,44]. Either of these endocrine dysfunctions would affect metabolism and has a strong impact on weight gain and obesity. Since our data suggested that a higher BMI z-score during longitudinal follow-up was associated with a higher risk of OSAS, we recommended regular thyroid and adrenal function evaluation, and if necessary, hormone replacement therapy for controlling or even lowering BMI.

In our cohort, patients with the nondeletion genotype had a lower OAHI. Previous studies revealed several phenotype differences in the nondeletion genotype of PWS, including less hypopigmentation [45] and increased incidence of psychosis [46]; however, different severities of SDB in nondeletion PWS were never reported. Because only five patients with nondeletion PWS were included in our cohort, we cannot make a definite conclusion that we can have different strategies to treat SDB in patients with deletion or nondeletion PWS, and hence a larger-scale study is required.

This study has several limitations. First, our study had the inherent weakness of a retrospective design and a relatively small sample size. Craniofacial structures, such as tonsillar size, muscle tone, and allergic rhinitis, are important confounding factors for OSA [47], which were not evaluated in this study due to the retrospective design. Larger, prospectively controlled studies are warranted to further evaluate the progression of OSA in patients with PWS. Second, in our sleep center, we used a thermistor, not a nasal pressure transducer, to detect air flow. This might lead to an underestimation of the severity of OSA. Third, we lacked information regarding standardized OSAS symptom scores, which revealed a high discrepancy to OAHI changes [48]. Despite these limitations, to the best of our knowledge, this is the first study to explore the progression of OSAS for up to 4 years in patients with PWS. We have provided evidence that natural resolution of OSAS in these patients is rare and watchful waiting may not be the best strategy.

In conclusion, SDB is common in patients with PWS. Natural resolution of OSAS was rare during the longitudinal follow-up, which suggests that watchful waiting may not be the best strategy. Patients with PWS with obesity and those with the deletion genotype tended to have more significant OSAS. Weight maintenance and careful selection of surgical candidates are important for OSAS treatment in patients with PWS.

## Figures and Tables

**Figure 1 children-09-00912-f001:**
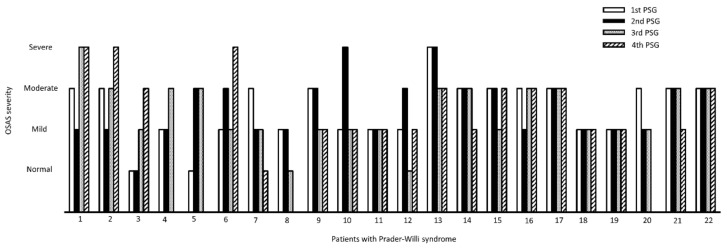
The illustration of OSAS severity in patients of Prader–Willi syndrome. OSAS severity varied in some patients, but it rarely resolved spontaneously during 4-year longitudinal follow-up.

**Table 1 children-09-00912-t001:** Demographic data and summary of the polysomnographic variables during longitudinal follow-up of patients with PWS for 4 years.

	1st PSG	2nd PSG	3rd PSG	4th PSG
*N*	22	22	22	18
Age (years)	11.7 ± 3.9	13.1 ± 4.1	14.4 ± 4.3	15.8 ± 4.5
Female (%)	12 (54.5)	12 (54.5)	12 (54.5)	10 (55.6)
Del/non-Del	17/5	17/5	17/5	15/3
BMI	22.8 ± 4.5	23.1 ± 4.1	23.8 ± 4.5	25.9 ± 6.4
BMI z-score	1.7 ± 1.3	1.6 ± 1.3	1.7 ± 1.5	2.3 ± 2.2
GH Tx (%)	18 (81.8)	18 (81.8)	18 (81.8)	16 (88.9)
Total sleep time (m)	414.6 ± 69.4	421.4 ± 52.2	393.0 ± 100.2	388.7 ± 73.6
Sleep efficiency (%)	87.4 ± 6.5	83.9 ± 8.3	79.8 ± 18.0	81.0 ± 12.2
Awake (%)	10.2 ± 6.1	12.5 ± 8.3	18.9 ± 16.7	17.6 ± 11.0
Stage 1 (%)	8.3 ± 4.2	7.6 ± 4.0	8.0 ± 5.5	9.5 ± 6.1
Stage 2 (%)	47.0 ± 11.8	45.1 ± 12.5	45.4 ± 16.5	46.6 ± 9.4
SWS (%)	23.2 ± 11.7	26.4 ± 11.6	24.6 ± 12.0	22.3 ± 10.3
REM (%)	20.4 ± 7.1	20.8 ± 5.9	21.9 ± 6.5	21.5 ± 4.8
CAI (/h)	0.3 ± 0.5	0.1 ± 0.2	0.0 ± 0.1	0.1 ± 0.1
OAHI (/h)	5.7 ± 6.4	6.5 ± 6.8	5.0 ± 3.5	6.4 ± 5.7
Oxygen concentration nadir (%)	83.1 ± 6.8	83.3 ± 6.5	82.8 ± 5.2	81.4 ± 9.8
Arousal index (/h)	12.5 ± 9.7	10.7 ± 7.6	8.0 ± 4.3	8.8 ± 6.1
No OSAS (%)	2 (9.1)	1 (4.5)	2 (9.1)	1 (5.6)
Mild OSAS (%)	8 (36.4)	10 (45.5)	10 (45.5)	7 (38.9)
Moderate OSAS (%)	11 (50)	9 (40.9)	8 (36.4)	7 (38.9)
Severe OSAS (%)	1 (4.5)	2 (9.1)	2 (9.1)	3 (16.7)

Values are expressed as the mean ± SD; Abbreviations: BMI, body mass index; CAI, central apnea index; Del, deletion; GH Tx, growth hormone treatment; OAHI, obstructive apnea–hypopnea index; OSAS, obstructive sleep apnea syndrome; PSG, polysomnography; PWS, Prader–Willi syndrome; REM, rapid eye movement; SWS, slow-wave sleep.

**Table 2 children-09-00912-t002:** Regression coefficients of OAHI in males and females with PWS according to the GEE model.

Parameter	Estimate	Wald χ^2^	*p*-Value
Intercept	−7.602	0.829	0.363
Sex ^a^	2.294	0.282	0.595
Age	0.116	0.062	0.803
BMI z-score	4.883	16.852	<0.001 **
PSG Time	0.259	0.101	0.751
Genotype ^b^	−3.728	4.086	0.043 *
GH Tx ^c^	4.886	1.511	0.219
BMI z-score × PSG Time	−0.476	3.585	0.058
Genotype × PSG Time	0.478	0.158	0.691

^a^ male = 0, females = 1; ^b^ deletion = 0, non-deletion = 1; ^c^ not using growth hormone = 0, using growth hormone = 1, * *p* < 0.05, ** *p* < 0.01. Abbreviations: BMI, body mass index; GEE, generalized estimating equation; GH Tx, growth hormone treatment; OAHI, obstructive apnea–hypopnea index; PWS, Prader–Willi syndrome; PSG, polysomnography.

## Data Availability

The data that support the findings of this study are available on request from the corresponding author, L.-P.T. The data are not publicly available due to their containing information that could compromise the privacy of research participants.

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
