# Peer review of "Progression of Obstructive Sleep Apnea Syndrome in Pediatric Patients with Prader–Willi Syndrome"

_children, 2022, doi:10.3390/children9060912_

Round 1
Reviewer 1 Report
This paper appears to be a potentially valuable detailed investigation of the natural history of sleep-disordered breathing in patients with PWS. Overall, the manuscript is a well-written paper that provides novel and interesting data. I have a few relatively minor comments, explained below.
1) The authors stated that Growth Hormone was administered to almost all study subjects and nevertheless most of the subjects showed BMI z-score increments in the 4-year duration. Pediatric patients with Prader-Willi Syndrome do not have impaired functions in the endocrine system limited to only GH. Studies have shown that PWS patients have impaired thyroid hormone secretion, leading to hypothyroidism. As wide known, hypothyroidism has a very strong impact on weight gain and obesity. Adrenal insufficiency is also another endocrine hormone secretion impairment found in PWS patients. The adrenal insufficiency also affects the metabolism regulation which again impacts weight gain and obesity. Insulin resistance is also another important metabolic factor that is found in PWS patients. Insulin resistance is widely known to impair blood sugar regulation and lead to weight gain, metabolic syndrome, and obesity.
If the authors suggest that BMI (or body weight) predominantly affects the AHI in PWS patients, the authors need to carefully monitor other metabolic-related endocrine hormones as well, and if necessary, administer hormone replacement therapy in order to control and assist in lowering BMI. Did the authors monitor other endocrine hormone levels?
Please add the above points to the discussion part.
2) Also discuss the reasons for deterioration of disordered breathing in patients who were normal at 1st PSG despite being diagnosed with OSA at a subsequent PSG (Patient number 3, 5).
Reviewer 2 Report
Dear Authors,
your paper is very interesting and original.
I have some questions.
Could you please explain better what do you mean with the expression
"oral appliance therapy"?
Do you mean MAD or ortho?
Speaking about surgical treatment, you mentioned adenotonsillectomy not orthognatic surgery. Could you please address better indications of the two surgeries?
Then could you explain better why the most of your patients detected for OSA were not be treated?
"obesity is a proved risk factor for OSAS [33, 223 34], although the severity of OSAS did not correlate positively with BMI" explain me better
Did you notice bruxism in your patients?
Very interesting your evolution about limitations of the study.
We hope to see new data very soon.
